# An Engineered Maturation Cleavage Provides a Recombinant Mimic of Foot-and-Mouth Disease Virus Capsid Assembly-Disassembly

**DOI:** 10.3390/life11060500

**Published:** 2021-05-29

**Authors:** Joseph Newman, David J. Rowlands, Tobias J. Tuthill

**Affiliations:** 1The Pirbright Institute, Pirbright GU24 0NF, UK; joseph.newman@pirbright.ac.uk; 2School of Molecular and Cellular Biology & Astbury Centre for Structural Molecular Biology, Faculty of Biological Sciences, University of Leeds, Leeds LS2 9JT, UK; d.j.rowlands@leeds.ac.uk

**Keywords:** foot-and-mouth disease virus, picornavirus, capsid assembly, antigenicity

## Abstract

Picornavirus capsids are assembled from 60 copies of a capsid precursor via a pentameric assembly intermediate or ‘pentamer’. Upon completion of virion assembly, a maturation event induces a final cleavage of the capsid precursor to create the capsid protein VP4, which is essential for capsid stability and entry into new cells. For the picornavirus foot-and-mouth disease virus (FMDV), intact capsids are temperature and acid-labile and can disassemble into pentamers. During disassembly, capsid protein VP4 is lost, presumably altering the structure and properties of the resulting pentamers. The purpose of this study was to compare the characteristics of recombinant “assembly” and “disassembly” pentamers. We generated recombinant versions of these different pentamers containing an engineered cleavage site to mimic the maturation cleavage. We compared the sedimentation and antigenic characteristics of these pentamers using sucrose density gradients and reactivity with an antibody panel. Pentamers mimicking the assembly pathway sedimented faster than those on the disassembly pathway suggesting that for FMDV, in common with other picornaviruses, assembly pentamers sediment at 14S whereas only pentamers on the disassembly pathway sediment at 12S. The reactivity with anti-VP4 antibodies was reduced for the 12S pentamers, consistent with the predicted loss of VP4. Reactivity with other antibodies was similar for both pentamers suggesting that major antigenic features may be preserved between the VP4 containing assembly pentamers and the disassembly pentamers lacking VP4.

## 1. Introduction

The *Picornaviridae* are a family of viruses with small (approximately 30 nm) non-enveloped icosahedral capsids containing a single copy of positive sense RNA genome. The family contains many important pathogens of humans and animals including poliovirus, human rhinovirus and foot-and-mouth disease (FMD) virus (FMDV). The global burden of FMD on livestock is significant and can incur large economic costs due to production losses, trade restrictions and control strategies including vaccination [1]. 

The FMDV capsid assembles via multimerisation of precursor subunits which encapsidate a single copy of the viral RNA genome. The viral RNA is infectious and when it enters the cytoplasm of cells, a single open reading frame (ORF) is translated into a polyprotein containing both structural (capsid precursor) and non-structural proteins [2]. During translation the capsid precursor, P1-2A, is released from the polyprotein by cleavage at its N-terminus by the viral Leader protease [3] and by a ribosomal slippage at the C-terminus of 2A, which separates P1-2A from the downstream polyprotein [4]. The N-terminus of P1-2A is myristoylated by a host-enzyme [5] and the viral 3C protease (3C^pro^) cleaves P1-2A in three places to generate the complex of VP0-VP3-VP1 which remains associated as a capsid protomer [6] and multimerises into a pentameric subunit. Twelve pentamers with intact VP0 are thought to multimerise in a concentration dependent manner to form provirions when in the presence of viral RNA [7] or empty capsids in the absence of RNA [8,9]. 

The cleavage of VP0 into VP4 and VP2, known as the “maturation cleavage” is the final event in picornavirus morphogenesis and is thought to be catalysed by RNA during encapsidation in poliovirus [10,11], however, empty capsids of FMDV can also contain cleaved VP0 suggesting a different mechanism [12,13,14]. Upon cleavage, VP4 and VP2 remain associated with the virion as individual capsid proteins with their now free termini participating in inter-protomer interactions [15] allowing the virus to adopt a “meta-stable” state for the infection of new cells [16]. 

During entry into host cells, FMDV virions rapidly disassemble into pentamers when the pH of endosomes becomes acidic (<pH 6.8) [17,18]. It has been proposed that the propensity of FMDV to fall apart into pentamers under acidic pH or elevated temperatures can be explained by the protonation of two histidine residues near the two-fold pentamer interface causing electrostatic repulsion between two pentameric subunits in increasingly acidic conditions [12,19,20]. Upon conversion of virions into pentamers, VP4 is lost and so pentamers disassembled from virions consist of VP2-VP3-VP1 [18]. 

Conventional FMD vaccines are commonly generated by inactivation with binary ethylenimine (BEI) and purification of virus particles. However, such preparations are unstable over prolonged storage or when cold chains are not maintained [21,22]. This loss of stability refers to the disassembly of virions into pentamers [23,24,25] and this correlates with loss of vaccine efficacy as the disassembled pentamers are no longer immunogenic [24,26,27,28]. Empty capsids lacking RNA, which form spontaneously during infection or which can be made recombinantly, are immunogenic [24,28] but are also unstable unless artificially stabilised [29]. 

The generation of native assembly pentamers is generally achieved by purifying VP0-containing empty particles and dissociating at high pH into pentamers. This is technically challenging and not possible for many viruses which either do not assemble a large proportion of empty capsids or do not retain a majority of uncleaved VP0. In this study we wished to generate and characterise recombinant FMDV capsid pentamers mimicking “assembly” and “disassembly” pentamers where the presence of VP4 could be controlled by an engineered cleavage site to mimic the maturation cleavage (Figure 1). The recombinant approach we describe bypasses the technical difficulties in production of native assembly pentamers and allows the direct comparison of pentamers which differ only in the cleavage status of VP0. We show that recombinant pentamers with intact VP0 (presence of VP4) have faster sedimentation than those with cleaved VP0, consistent with 14S and 12S sedimentation phenotypes. Pentamers with cleaved VP0 had reduced reactivity with anti-VP4 antibodies, confirming the loss of VP4 but reactivity with other antibodies was similar for both pentamers. 

## 2. Materials and Methods

### 2.1. Generation of Plasmids

The plasmids used in this work were generated according to standard molecular biology protocols [30]. The wild-type (*wt*) version of FMDV capsid precursor was expressed from an existing plasmid (pBG200-P1-Δ2A) [31] which encoded P1 and a truncated form of 2A capsid precursor from the O1 Manisa strain of FMDV. 

Overlap PCR mutagenesis was used to generate a mutant of the capsid precursor that had an independently cleavable VP4/VP2 junction. Two sets of primers (5′-CTGAGGCTTCTTTAAAAGCGCTC and 5′-GGGCCCCTGGAACAGAACTTCCAGAAGACCGCTGAAAGCGGAAC, 5′-CTGGAAGTTCTGTTCCAGGGG-CCCAAAACCGAGGAGACCACTCTTCTG and 5′-CTGTGTCTCGCCACCGTAATTCT where red bases indicate mutated primer tails), were used to generate two amplicons using KOD polymerase (Roche). These two amplicons were purified by agarose gel electrophoresis and gel extracted using GFX PCR DNA and gel band purification kit (GE Healthcare, Chicago, IL, USA), before being added in equi-molar quantities to a second PCR reaction using only the flanking primers from above to generate a single amplicon which was ligated (T4 ligase; NEB, Ipswich, UK) back into the pBG200-P1-Δ2A vector. The mutated sequences were confirmed by sequencing.

### 2.2. Proteases

FMDV 3C^pro^ was expressed in E.coli from plasmids obtained from Stephen Curry at Imperial College London [32] according to the method we have described previously [31]. Purification of 3C^pro^ was via a C-terminal His^6^ tag using immobilised metal affinity chromatography (HisTrap FF; GE Healthcare). The purified enzyme was dialysed (10K MWCO, Slide-a-Lyzer; Pierce, Rockford, IL, USA) against 50 mM HEPES pH7.1, 0.2 M NaCl, 1 mM EDTA, 1 mM β-mercaptoethanol and 5% glycerol.

Precission protease (Pre^pro^, Novagen, Madison, WI, USA) was obtained commercially. 

### 2.3. Cell Free Processing and Assembly

Radiolabelled capsid precursors and pentamers were generated according to a previous protocol [31]. Briefly, expression plasmids (20 ng/µL) were programmed into rabbit reticulocyte lysates (TnT quick; Promega, Madison, WI, USA) containing 8 Bq/µL L-{^35^S]-Methionine (EasyTag; Perkin Elmer, Waltham, MA, USA) according to the manufacturer’s instructions to generate capsid precursors. These reactions were then mock treated with PBS (without cations; Pirbright CSU, Woking, UK) or treated with 1 µM 3C^pro^ or 0.2 U/µL Pre^pro^ and incubated for 1 h at 37 °C, or reactions were treated sequentially with each protease for a second hour at 37 °C. After incubation, free radiolabel was removed from the reaction mixtures by three rounds of dialysis (10 kDa Slide-A-Lyzer mini dialysis cassettes pre-blocked with 1% bovine serum albumin in PBS) against 500 mL PBS. 

### 2.4. SDS-PAGE and Fluorography

To analyse processing and assembly reactions, proteins were resolved through 12% Tris-glycine SDS-PAGE gels (mini-protean II; Bio-rad; 37.5:1 Acrylamide/Bis acrylamide), the gels were soaked in 1 M sodium salicylate (VWR) for 30 min, dried for one hour at 80 °C (DrygelSR, Hoefer, San Francisco, CA, USA), exposed to film at −80 °C overnight and developed using standard photographic reagents.

### 2.5. SDGs

The sedimentation of capsid precursors, protomers and pentamers was analysed by sucrose density gradients (SGDs) by the method described by a previous protocol [31]. Briefly, samples were sedimented through 5 mL 5–30% sucrose gradients (Gradient master; Biocomp) for 6 h at 286,794× *g* average, at 10 °C in an SW55Ti rotor (Beckman Coulter, Brea, CA, USA).

Gradients were fractionated (Piston gradient fractionator, Biocomp, Fredericton, Canada) into 24 equal fractions which were diluted 1:5 in scintillation fluid (Optiphase Supermix; PerkinElmer) for counting (LS6500 multipurpose scintillation counter; Beckman Coulter). 

### 2.6. Purifying Antibodies from Bovine Sera

To generate antibody reagents specifically reactive with VP4 and VP2, two peptides were synthesised (Peptide Protein Research) that corresponded to the highly conserved N-terminal 45 amino acids of FMDV VP2 (DKKTEETTLLEDRILTTRNGHTTSTTQSSVGVTYGYATAEDFVSGKKKKKK-biotin) and VP4 (Myristic acid-GAGQSSPATGSQNQSGNTGSIINNYYMQQYQNSMDTQLGDNAISGKKKKKK-biotin) with lysine residues at the C-terminus for enhanced solubility followed by biotin for purification. The peptides were bound to StrepTrap HP columns (GE Healthcare) and unbound peptide was removed by washing. Bovine sera raised against FMDV (WRLFMD Pirbight), was clarified by low speed centrifugation and run through the column using a peristaltic pump (EP1 Econo pump; Bio-Rad, Hercules, CA, USA). The columns were then washed with buffers of increasing acidity (50 mM glycine, 150 mM NaCl pH values were 6, 5, 4, 2.7 and 1.9). Each fraction eluted from the column was neutralised with 10× neutralisation buffer (1 M Tris-HCl, pH 8; 1.5 M NaCl, 1 mM EDTA). The presence of antibody in eluted fractions was confirmed by SDS-PAGE. 

### 2.7. Pentamer Immunoprecipitation

Peak gradient fractions containing radiolabelled pentamers were pooled and dialysed (G2 3 mL 20kDa MWCO dialysis cassettes; ThermoFisher) against 1 L PBS twice to remove sucrose. Radioactivity in each sample was counted. Samples of pentamers (approx. 10^5^
^35^S counts) were mixed with an amount of antibody pre-determined to give complete immunoprecipitation of *wt* pentamers. Antibody-pentamer mixtures were incubated for two hours with shaking at room temperature. Next, 50 µL protein A magnetic beads for guinea pig polyclonal antibodies or protein G magnetic beads (Thermo Fisher Scientific, Waltham, MA, USA) for bovine and mouse antibodies were washed and then incubated with the antibody-pentamer complexes with shaking for one hour at room temperature. The complexes were washed three times with PBS-T (0.1% Tween-20; Sigma, St. Louis), and then any immunoprecipitated protein was eluted with 50 mM glycine pH 1.9 in PBS-T. Samples were read by scintillation counting as above and each pull down was expressed as proportion of the input radioactivity. 

## 3. Results

A final event in picornavirus morphogenesis is the cleavage of VP0 into VP4 and VP2 (the maturation cleavage). 

Previously we have shown that a cell-free system can be used to study FMDV capsid precursor processing and pentamer assembly [31,33] and here we have further developed this system to analyse the effect of the maturation cleavage on pentamer properties. 

### 3.1. An Artificial Maturation Cleavage Site Can Be Independently Processed

The amino acid sequence at the FMDV VP4/2 junction is highly conserved between FMDV serotypes; LFGALLA/DKKTEETT (where “/” represents the protein junction; [34,35]). We modified an existing FMDV serotype O capsid precursor expression plasmid to insert a PreScission Protease (Pre^pro^) cleavage site at this junction; LLEVLFQ/GPKTEET (termed *mat*, for maturation mutant). 

The Pre^pro^ protease is a recombinant commercial version of purified human rhinovirus 14 3C^pro^. Human rhinovirus 3C^pro^ has fairly stringent requirements for Q/G at P1/P1′ sites of cleavage junctions whereas FMDV 3C^pro^ is more tolerant in its amino acid requirements at cleavage junctions [36] due to an extended β-ribbon structure in the substrate binding cleft [37]. 

To confirm that the engineered maturation cleavage site at the VP4/2 boundary was accessible to Pre^pro^, and that processing of the FMDV 3C^pro^ cleavage sites and the maturation site could be controlled independently, *wt* (Figure 2, upper panel) or *mat* (Figure 2, lower panel) capsid precursors were expressed and cleaved by each protease individually or by both proteases sequentially. In uncleaved *wt* samples, a band corresponding to the expected size of P1-2A was observed (Figure 2, upper panel, Lane 1), which was cleaved by FMDV 3C^pro^ to produce bands corresponding to the expected sizes of capsid proteins VP0, VP1 and VP3 (Figure 2 upper panel, Lane 2). No further changes were observed following subsequent or prior treatment with Pre^pro^ (Figure 2, upper panel, Lanes 3–5). In contrast, VP0 from the *mat* P1-2A was replaced by VP2 as expected from the additional cleavage event (Figure 2, lower panel, Lanes 3 and 5). VP2 and VP3 co-migrate as a single band of slightly larger apparent molecular weight than VP3 alone. Pretreatment of *wt* with Pre^pro^ did not affect the precursor or product profiles but pretreatment of *mat* P1-2A produced a band corresponding to VP2-VP3-VP1 (Figure 2, Lane 4) as expected. Subsequent treatment with 3C^pro^ produced the same banding pattern as when the two proteases were used in the alternative order (Figure 2, lower panel, Lane 5). The relative intensities of the bands seen in the autoradiographs reflect the different methionine contents of the O1 Manisa capsid proteins—VP1,VP2, VP3, VP0, (2, 4, 4 and 6 methionine residues respectively). VP4 (7kDa) was too small to be resolved on these gels. 

These results demonstrated that Pre^pro^ cleaved specifically at the introduced site in the capsid precursor P1-2A independently of FMDV 3C^pro^.

### 3.2. Maturation Site-Cleaved Pentamers Have a Slower Sedimentation Coefficient Than Assembly Pentamers

We have shown previously that incubation of recombinant P1-2A with 3C^pro^ resulted in the assembly of pentamers [31,33]. We now wished to use the engineered maturation site to analyse the effect of VP0 cleavage on sedimentation of protomers and assembled pentamers. P1-2A precursors (*wt* or mat) were treated with proteases or mock treated and the sedimentation of the resulting assembly products was determined by centrifugation through sucrose density gradients. 

Mock treated *wt* P1-2A sedimented as a single peak at the expected position for capsid precursor (5S) (Figure 3; upper panel blue trace) and treatment with 3C^pro^ produced two peaks sedimenting at the expected positions for 5S protomers and assembled pentamers (Figure 3; upper panel red trace). Treatment of mat P1-2A with 3C^pro^ alone produced similar results as with *wt* P1-2A (Figure 3; lower panel blue and red traces respectively). However, following additional treatment with Pre^pro^, the sedimentation rates of both the protomer and pentamer peaks were reduced (Figure 3; lower panel orange trace). In this case, the pentamer peak sedimented at a position identical to disassembly pentamers derived by denaturation of virus particles, which have a sedimentation coefficient of 12S [31]. Sedimentation of the protomer was similarly reduced. Interestingly, initial treatment with Pre^pro^ followed by 3C^pro^ resulted again in a slower sedimenting protomer but in this case no peak corresponding to pentamers was seen (Figure 2; lower panel, black trace), suggesting that the presence of uncleaved VP0 is essential for higher order assembly to occur.

These results demonstrated that the cleavage of VP0 slightly reduced the sedimentation coefficients of both protomer and pentamer particles, suggesting that the multimeric state of these pentamers was still intact and that VP0 cleavage only minimally altered the structure. The sedimentation coefficient of *wt* pentamers was estimated to be 14S, consistent with the assembly competent pentamers of other picornaviruses [2], whereas cleavage of VP0 reduced this value to 12S, the same as seen for pentamers released from virus particles by thermal or acid denaturation. 

### 3.3. Maturation Site-Cleaved Pentamers and Wild-Type Pentamers Are Recognised Similarly by Anti-FMDV

The sedimentation analysis above demonstrated that VP0-cleaved *mat* pentamers were structurally different to *wt* pentamers. To determine whether this change in structure affected the antigenic characteristics of the pentamers, a panel of anti-FMDV antibodies was used to immunoprecipitate purified *wt* and VP0-cleaved *mat* pentamers. The antibodies used and their reactivities are listed in Table 1.

VP4-specific antibodies, affinity purified from FMDV-infected bovine sera using a peptide corresponding to the N-terminal 45 amino acids of VP4, were used to demonstrate that the VP0 maturation cleavage had released VP4 from the *mat* pentamers. These VP4-reactive antibodies efficiently immunoprecipitated *wt* pentamers but, in contrast, only a minor proportion of VP0-cleaved *mat* pentamers were immunoprecipitated (Figure 4). This was consistent with the predicted loss of the majority of VP4 after cleavage of *mat* pentamers by Pre^pro^ and suggested that the VP0-cleaved *mat* pentamers were equivalent to disassembly pentamers which lack VP4. VP2-specific antibodies were affinity purified in the same way as for the VP4-speciific antibodies. These antibodies reacted equally well with both *wt* and VP0-cleaved *mat* pentamers indicating that the VP0 cleavage and loss of VP4 had not affected the binding of antibodies to VP2. 

Polyclonal sera against whole FMDV immunoprecipitated the *wt* and VP0-cleaved *mat* pentamers equally well (Figure 4). A monoclonal antibody (D9) specific for the GH-loop of FMDV, which includes a major immunogenic linear epitope, reacted equally with both pentamers. A second monoclonal antibody, IB11, with an undefined conformational epitope on the capsid also recognised both particles. Together, these data show that apart from the expected reduction in reactivity of VP4-specific antibodies, both intact VP0 containing *wt* pentamers and VP0-cleaved *mat* pentamers were recognised similarly by a number of virus specific polyclonal and monoclonal antibodies.

## 4. Discussion

In this study, we have used a recombinant system for mimicking the maturation cleavage in FMDV pentamers. We have shown that this introduced cleavage site can be cleaved independently of the 3C^pro^ processing sites and that upon cleavage the pentamers undergo a change in sedimentation. This change is consistent with a switch from the 14S assembly pentamer described in most other picornaviruses [2,40] and the 12S disassembly form of pentamer which is most often described in relation to FMDV. Historically, both assembly and disassembly pentamers in FMDV are referred to as having a 12S sedimentation in sucrose [28]. For FMDV, the 12S disassembly form of the pentamer is the easier of the two forms to study and is also arguably more significant due to the implication of disassembly for vaccines. Consequently, assembly pentamers have been overlooked and both forms of FMDV pentamers have been indiscriminately labelled as 12S. In contrast, other well-studied viruses in the enterovirus genus of the family do not disassemble and remain as an intact capsid structure, meaning that only 14S assembly pentamers have been available for characterisation [2,41]. Therefore, the current study reconciles the capsid assembly of FMDV with other picornaviruses. 

The data presented here suggest that the loss of VP4 from disassembly pentamers results in a less hydro-dynamically compact structure that sediments more slowly in sucrose gradients relative to assembly pentamers. This also appears to be true of protomers lacking VP4. The structure of pentamers dissociated from FMDV has been previously determined [42] and in addition to lacking VP4, such disassembly pentamers appeared structurally relaxed compared to pentamers observed in intact virions, with disordered VP1 and VP2 N-termini and a rotation in the VP3 protein resulting in a less compact structure consistent with slower rate of sedimentation.

In existing structures of the FMDV virion, the VP4/2 site in VP0 is internal in the capsid [15] and therefore inaccessible to external proteases, which has resulted in a hypothesis that the cleavage involves an RNA-induced auto-catalytic event [11]. This however was not true for the recombinant pentamers in the current study and Pre^pro^, which is the 3C^pro^ from HRV14, was able to access and cleave at this site. This suggests that the VP4/2 junction of VP0 in *wt* pentamers is exposed to the cellular environment and has the potential to be cleaved. The mechanism of maturation cleavage in FMDV is inferred from studies with poliovirus where the RNA has an autocatalytic role upon encapsidation but the presence of FMDV empty capsids with cleaved VP0 [13,14] suggests a different mechanism in FMDV and that perhaps the accessibility of this site makes it vulnerable to cleavage. The Pre^Pro^ (3C^pro^) recognition sequence in enteroviruses rarely varies at the P1/P1′ position; being a Q/G pair [36]. Some FMDV cleavage junctions are also Q/G pairs but FMDV 3C^pro^ is more tolerant in all positions (P4 to P4′) [32,34,41,43] around the cleavage junction. Therefore, it seems unusual that FMDV 3C^pro^ cannot cleave this site in the *mat* pentamers and this may warrant further investigation of cleavage site specificity. 

We also showed that the 12S form of pentamer had a reduced reactivity with anti-VP4 antibodies after VP0 cleavage, consistent with loss of VP4 from the pentamer. Despite a measurable change in conformation of the pentamer and apparent loss of VP4 upon cleavage of VP0, the overall reactivity of the pentamer with other antibodies was unchanged, suggesting that the global antigenicity of assembly and disassembly pentamers may remain similar. 

Earlier studies showed that FMDV pentamers from the assembly pathway could generate a protective immune response in animals in contrast to dissociated pentamers which were not protective [28]. This suggested that the two forms of pentamer were antigenically distinct. However, in the current study, we were not able to differentiate between the antigenicity of the recombinant 14S assembly and 12S disassembly pentamers when using a panel of monoclonal and polyclonal antibodies, with the only apparent difference being the loss of reactivity with anti-VP4 antibodies. Although VP4 is an internal capsid protein, it has been shown in other picornaviruses to be transiently externalised in a process known as capsid breathing [44,45], so that antibodies raised against the VP4 protein of HRV and PV can neutralise these viruses [45,46,47]. It is therefore possible that VP4 specific antibodies may also play a role in the response to FMDV. It should also be noted that studies of pentamer antigenicity cannot include epitopes formed at the pentamer interfaces of fully assembled capsids, such epitopes may also be important.

The work presented here demonstrates a recombinant system to investigate differences in the properties of picornavirus capsid components before and after the maturation cleavage, which may be useful in further studies of capsid assembly and vaccine antigenicity. 

## Figures and Tables

**Figure 1 life-11-00500-f001:**
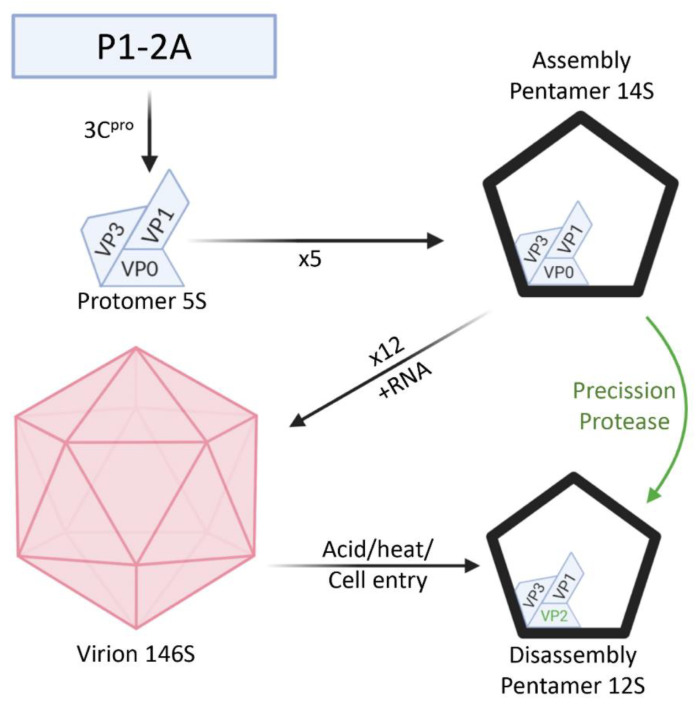
Assembly and disassembly of FMDV capsids and control of pentamer switching. Black arrows indicate pathway of normal FMDV capsid assembly and disassembly. The green arrow represents the pathway taken by the ‘maturation engineered’ pentamers generated in this study. Processing of P1-2A facilitates folding of P1-2A into an assembly competent protomer (5S) followed by assembly of 5 protomers into a pentameric structure with 14S sedimentation containing intact VP0 protein. Twelve pentamers assemble into intact capsids in a concentration dependent manner and if viral RNA is encapsidated they form virions (146S). Low pH, heat or cell entry trigger the disassembly of intact capsids into disassembly versions of pentamers (12S) containing VP2 and lacking VP4. An introduced maturation cleavage mutation allows conversion of assembly pentamers directly into disassembly pentamers. Created with Biorender.com.

**Figure 2 life-11-00500-f002:**
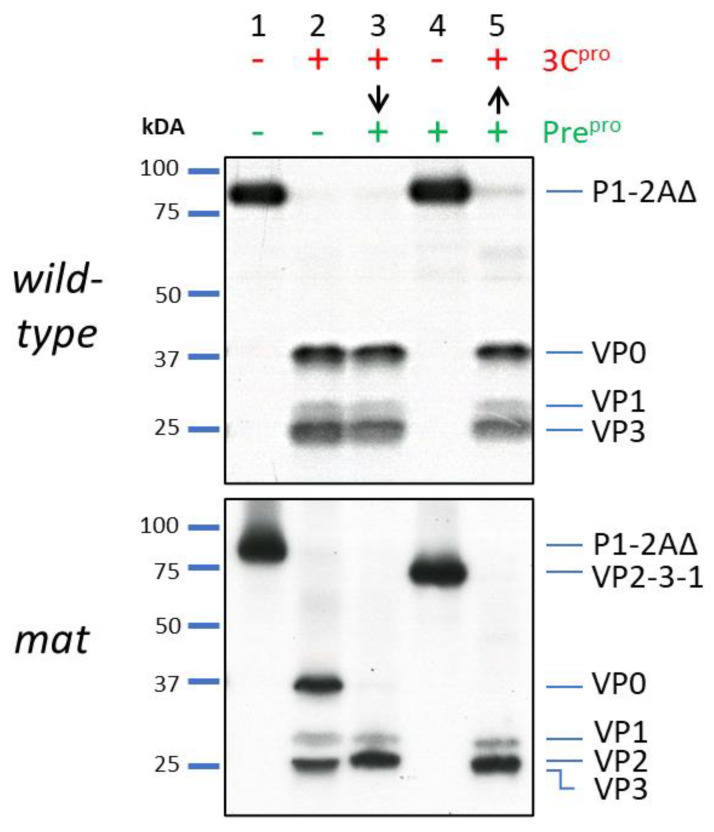
FMDV capsid precursor mat mutant can be cleaved by Pre^pro^ independently of 3C^pro^ processing. Cell free expression reactions of *wt* (top panel) or mat (bottom panel) were either unprocessed (Lane 1), processed with 3C^pro^ (Lane 2) and then Pre^pro^ (Lane 3), or processed with Pre^pro^ (Lane 4) and then 3C^pro^ (Lane 5) as indicated above the gel image for one hour at 37 °C. Arrows indicate sequential treatment with both proteases. Samples were resolved by SDS-PAGE and visualised by fluorography. The predicted sizes of capsid proteins and precursors are identified to the right of the gel images and the positions of molecular weight markers are indicated to the left.

**Figure 3 life-11-00500-f003:**
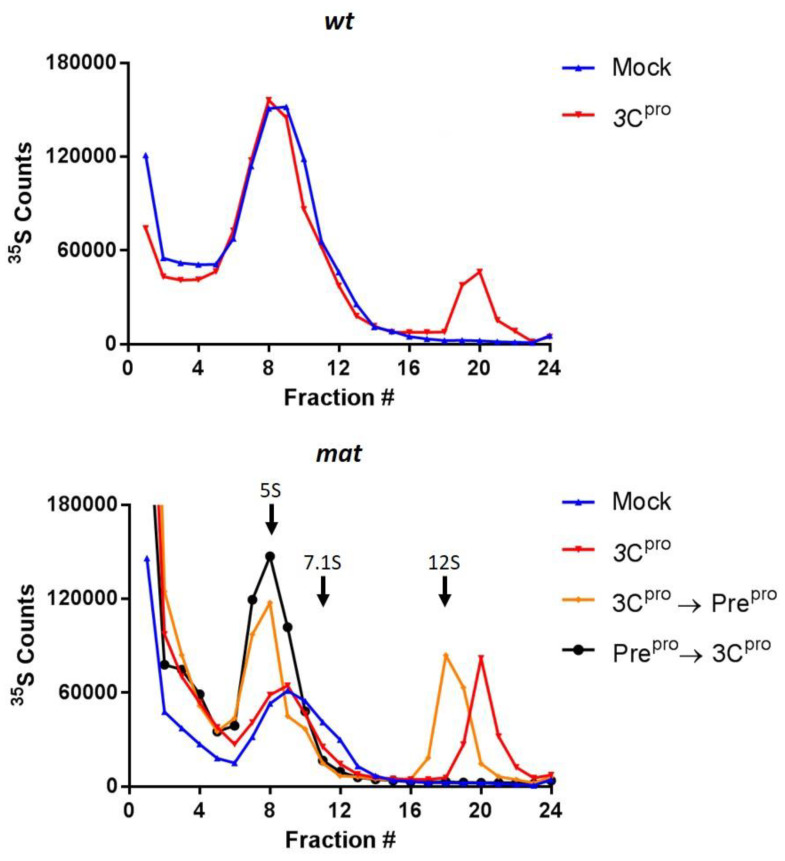
FMDV mat mutant pentamers and protomers sediment slower after cleavage. Cell free expression reactions of *wt* (upper panel) and mat (lower panel), were sedimented through sucrose density gradients and the radioactive counts were analysed in 24-equal fractions. Traces correspond to treatments as follows; mock processing (blue), 3C^pro^ (red), sequential treatment with 3C^pro^ then Pre^pro^ (orange) or Pre^pro^ then 3C^pro^ (black). Gradient profiles are single data points, representative of the position of pentamer peaks observed on multiple occasions. Arrows indicate peak position of sedimentation markers as determined in [31].

**Figure 4 life-11-00500-f004:**
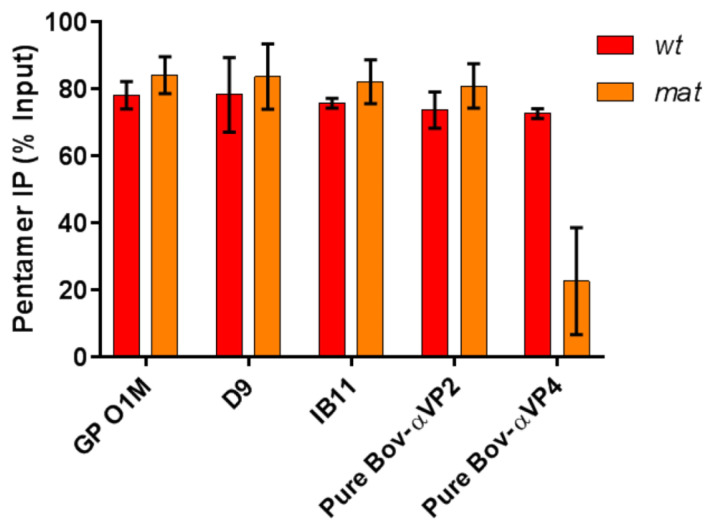
FMDV mat mutant pentamers lack VP4 but are antigenically indistinguishable from *wt* pentamers. Antibodies were incubated with either *wt* (red) and mat (orange) pentamers before the addition of either Protein A (GP sera) or Protein G (all other antibodies) magnetic beads. Radioactivity bound to the beads was read by scintillation counting. Pentamer immunoprecipitation is expressed as a proportion of input counts. The data shown are the mean of two independent biological repeats and error bars represent the range of the data.

**Table 1 life-11-00500-t001:** List of antibodies.

Antibody	Description	Reactivity	Reference
**Guniea pig (GP) α-O1M**	Reference sera from Pirbright FMD World Reference Laboratory raised in guinea pigs against FMDV strain O1M	Whole virus capsid	N/A
**D9**	Monoclonal antibody raised against FMDV type O	GH-loop of FMDV type O VP1—linear epitope	[38]
**IB11**	Monoclonal antibody raised against FMDV type O	Whole virus capsid—conformational epitope	[39]
**Purified bovine αVP2-N45**	Antibodies specific for the conserved N-terminus of VP2 affinity purified from FMDV α-SAT2 sera	VP2N45	Generated in this study
**Purified bovine αVP4-N45**	Antibodies specific for the conserved N-terminus of VP4 affinity purified from FMDV α-SAT2 sera	VP4N45	Generated in this study

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
