# Peer review of "An Engineered Maturation Cleavage Provides a Recombinant Mimic of Foot-and-Mouth Disease Virus Capsid Assembly-Disassembly"

_life, 2021, doi:10.3390/life11060500_

Round 1

Reviewer 1 Report

The manuscript by Newman et al. reports on a comparison between wildtype and mutant pentamers of FMDV, whereby in the latter VP0 can be processed in a controlled manner. The manuscript is overall well written and the approach would be of high interest for other picornaviruses, where no disassembly intermediates have been studied. The body of data is not very huge, nevertheless the manuscript is suitable for Life after minor revision.

Major comments:

No major difference in antigenicity is observed between cleaved and uncleaved pentamers. From the introduction, the reviewer expected something different, please clarify in intro.

What is the major advantage of the mutant besides being able to produce disassembly pentamers that can also be obtained from disassembled particles? This should already come across in the introduction and not only in discussion.

In absence of other factors, the wildtype is also in assembly state. What happens if the mat mutant assembles into particles would these then not mature that would be an important difference and experiment.

l.328 ff: this statement is not supported by the presented data. The cleavage side here is accessible in pentamers, whether that is the case for viral particles has not been studied here. The fact that the only difference in antigenicity was the loss of VP4 antibody binding in conjunction with results l.353 ff is maybe a stronger argument for the hypothesis of VP0 cleavage through cellular proteases. The authors should also pick up in the discussion, the point in l. 72 ff. The immunogenicity of empty capsids suggests that not (only) VP4 but the contact between pentamers is decisive.

Minor:

l.169 should read: "and then incubated"?

l.211: protein name to intensity is displayed in a confusing manner. Why is VP4 not resolved, isn't it reasonably different in size, rather too small or no methionine?

Fig.2: It would be useful to have some size indication next to gels.

Check throughout text and captions that pro in Prepro or 3Cpro appears as superscript.

l.281 specific

Reviewer 2 Report

This study compares some characteristics of pentameric intermediates that occur during either foot-and-mouth disease virus (FMDV) assembly or disassembly. As one consequence of a maturation step during FMDV morphogenesis, assembly pentamers and disassembly pentamers are different. The authors use an elegant approach to generate both types of pentamers and investigate differences and similarities in overall structure and antigenicity. The experiments have been competently performed, the results are clear and the main conclusions adequately justified. 

Author Response

No response required

Reviewer 3 Report

The paper by Newman and colleagues introduces a novel approach to study assembly and disassembly of picornavirus virions that can be useful for many researchers. Picornavirus capsid protein precursor is processed by the proteases 3C or 3CD into VP0, VP3 and VP1 capsid proteins that assemble into pentamers and further into procapsids and mature capsids. The final maturation cleavage of VP0 into VP4 and VP2 is believed to be autocatalytic in the case of enteroviruses, but may have a different mechanism in other picornaviruses. The authors replaced an authentic cleavage site in the VP0 capsid protein of FMDV for the one that can be recognized by an exogenously added protease with a specificity distinct from that of the FMDV 3C. This allowed them to fully control the capsid protein precursor processing and generate in vitro the “assembly” VP0-VP3-VP1 and “disassembly” VP2-VP3-VP1 pentamers. They further investigated their biophysical and immunological properties. The VP2-VP3-VP1 pentamer sedimented slower than the VP0-VP3-VP1 indicating that the loss of VP4 induces noticeable conformational changes but not the complete pentamer disassembly. Interestingly, the pentamers were not detected if VP0 was cleaved at the VP4-VP2 site before processing of the other cleavage sites, indicating that intact VP0 is required for pentamer assembly. The immunoreactivity of “assembly” and “disassembly” pentamers with either purified peptide-specific antibodies or polyclonal anti-FMDV serum was similar, except for antibodies against VP4, consistent with the loss of this peptide upon VP0 cleavage. Overall, the work is performed on a high technical level and the results are convincing and presented logically and concisely.

Author Response

No response required

Reviewer 4 Report

The manuscript entitled An engineered maturation cleavage provides a recombinant mimic of foot-and-mouth disease virus capsid assembly-disassembly describes the generation of a recombinant system that mimic the maturation cleavage in FMDV capsid pentamers. Namely, the authors idea is to control the presence of Vp4 by introducing an engineered cleavage site in FMDV pentamers that allows its cleavage independently from the 3Cpro processing sites.

The manuscript is well structured. I have only one question about SDS-PAGE analysis (figure 2). It is important to present an additional line for a protein standard that gives an indication of the molecular weight. Moreover, the composition of the gel and its percentage are important parameters that must be specified. Finally, in the legend of figure 2 it is necessary to clarify what each lane refers to.   

Furthermore, sometimes during the text “pro” after 3C and Pre it is quoted as superscript and other times not (examples: lines 226, 234, 236, 241, 242...). Are there any differences? If not, it should be better to make everyone uniform. 
